# Airway Epithelial Cell Junctions as Targets for Pathogens and Antimicrobial Therapy

**DOI:** 10.3390/pharmaceutics14122619

**Published:** 2022-11-27

**Authors:** Nannan Gao, Fariba Rezaee

**Affiliations:** 1Department of Inflammation and Immunity, Lerner Research Institute, Cleveland Clinic Foundation, Cleveland, OH 44195, USA; 2Center for Pediatric Pulmonary Medicine, Cleveland Clinic Children’s, Cleveland, OH 44195, USA

**Keywords:** airway epithelial cells, apical junctional complex, tight junction, adherens junction, bacterial infection, viral infection, epithelial barrier dysfunction, trans-epithelial electrical resistance, permeability, antimicrobial therapy

## Abstract

Intercellular contacts between epithelial cells are established and maintained by the apical junctional complexes (AJCs). AJCs conserve cell polarity and build epithelial barriers to pathogens, inhaled allergens, and environmental particles in the respiratory tract. AJCs consist of tight junctions (TJs) and adherens junctions (AJs), which play a key role in maintaining the integrity of the airway barrier. Emerging evidence has shown that different microorganisms cause airway barrier dysfunction by targeting TJ and AJ proteins. This review discusses the pathophysiologic mechanisms by which several microorganisms (bacteria and viruses) lead to the disruption of AJCs in airway epithelial cells. We present recent progress in understanding signaling pathways involved in the formation and regulation of cell junctions. We also summarize the potential chemical inhibitors and pharmacological approaches to restore the integrity of the airway epithelial barrier. Understanding the AJCs–pathogen interactions and mechanisms by which microorganisms target the AJC and impair barrier function may further help design therapeutic innovations to treat these infections.

## 1. Introduction

The airway epithelium is at the front line defending the human respiratory system against inhaled environmental insults such as viruses, bacteria, and allergens [1,2]. From the nasal cavity to the alveoli, specialized epithelial cells line the entire respiratory tract. There are distinct regional differences in the epithelial cell populations as the functions vary across different airway levels. The major epithelial cell types lining the airways are generally ciliated and secretory cells in adaptation to the mucociliary escalator, while the distal alveolar regions are lined with alveolar epithelial type I and type II cells to enable efficient air exchange [3]. As an essential part of the innate immune system, the airway epithelium is pivotal to the barrier function of the airway [4]. The primary contributors to the airway epithelial barrier function include mucociliary clearance, antimicrobial peptides secreted by the airway epithelial cells (AECs), and intercellular apical junctional complexes (AJCs) that regulate barrier integrity [5]. The AJCs are formed between adjacent epithelial cells on the apical lateral membranes and are composed of tight junctions (TJs) and adherens junctions (AJs) (Figure 1). Through interactions between the extracellular domains of the membrane-spanning adhesion molecules, TJs and AJs establish connections between neighboring cells. TJs are located at the apex of the lateral membranes between adjoining cells. TJs encircle the cell and form a “seal” between cells. In addition to this “seal”, TJs maintain the cell polarity by separating apical and basolateral membrane components and can control the paracellular permeability [6,7]. TJs comprise three families of transmembrane proteins: the claudin family (claudin-1 through 27 in mammals) [8], the tight junction-associated marvel protein (TAMPs) family (occludin, tricellulin, and MARVEL domain-containing protein 3), and the immunoglobulin family (junctional adhesion molecules (JAM-A) and coxsackie adenovirus receptor (CAR)) [1]. Inside the cell, TJs are connected to the cytoplasmic scaffolding proteins, such as zonula occludens (ZO-1, ZO-2, and ZO-3) and cingulin [9], and in turn, attached to the actin cytoskeleton [10,11,12,13]. AJs are more basally located than TJs and are crucial for initiating and maintaining cell–cell adhesion [14]. AJs are comprised of two primary transmembrane protein families: the cadherin family (such as E-cadherin) and the nectin family. Intracellularly, AJs bind to a cytoplasmic scaffolding and signaling complexes consisting of p120 catenin, β-catenin, and α-catenin proteins and are hence anchored to the actin cytoskeleton, forming a hub for signaling transduction and transcriptional regulation [1,6,7]. Microorganisms such as bacteria and viruses have evolved to utilize various mechanisms to breach the airway epithelial barrier as an early critical step to establish infection. Emerging evidence has revealed increased barrier permeability as a common characteristic for infected cells during respiratory bacterial and viral infection [15,16,17,18,19,20,21]. Increasing permeability not only allow bacteria and virus to reach tissues underneath the airway epithelium and bloodstream but will also facilitate secondary invasion of other allergens and pathogens [2]. Therefore, strategies to modify barrier permeability bear great potential in developing antimicrobial therapeutics.

Most human bacterial pathogens are capable of asymptomatic colonization in the respiratory tract, which can progress toward mild to severe infections such as sinusitis and pneumonia [22]. Bacteria penetrate the airway epithelium by damaging or crossing the AECs. Bacteria can cause lysis or apoptosis of the host cells by releasing exotoxins and enzymes [23]. On the other hand, bacterial infections could lead to disruption of the epithelial barrier structure and function via targeting TJs and AJs (Table 1). Respiratory viral infections result in diseases from the common cold to bronchiolitis and acute respiratory distress. Airway and alveolar epithelial cells are the primary targets for these inhaled pathogens, and respiratory viruses usually enter the host cells via binding to specific receptors [24]. Interestingly, components of the TJs can be hijacked by some viruses to aid their infectious cycle. For example, CARs are used by adenoviruses and coxsackievirus B as receptors during their entry processes [25]. While some respiratory viruses disrupt barrier dysfunction by causing cytotoxicity in host cells, others cause epithelial barrier disruption without affecting cell viability [2]. Increasing evidence has illustrated mechanisms utilized by respiratory viruses to disassemble TJs and AJs by regulating protein expression or localization of their components (Table 2).

In recent years, studies using airway epithelial cell lines, primary epithelial cells cultured at the air–liquid interface (ALI), and murine models have helped to identify the structural and functional damages of AJCs caused by microbial infection as well as the underlying mechanisms. Characterizing the mechanisms by which pathogens impair the AJCs and damage barrier function will help to improve therapeutic innovations in related disease areas. In this review, we will discuss *Staphylococcus aureus* (*S. aureus*), *Streptococcus pneumoniae* (*S. pneumoniae*), *Pseudomonas aeruginosa* (*P. aeruginosa*), *Burkholderia*, *Haemophilus influenzae* (*H. influenzae*), respiratory syncytial virus (RSV), human rhinovirus (HRV), influenza viruses, human parainfluenza viruses (HPIV), and coronavirus family (Figure 2). We will briefly summarize each microorganism’s clinical manifestation, health burdens, and challenges in infection management. Then, we will focus on the current knowledge regarding their impacts on airway epithelial barrier integrity and the pathophysiologic mechanisms of pathogen-induced disruption of the AJC. Finally, we will reflect on the present and potential approaches to restoring the epithelial barrier during infection.

## 2. Bacteria

### 2.1. Staphylococcus aureus

*Staphylococcus aureus* (*S. aureus*) is a Gram-positive bacterium that belongs to the *Micrococcaceae* family [79,80]. Humans are natural carriers of *S. aureus* [81], and these bacteria are commonly present in healthy individuals’ skin and mucous membranes, with the nasal cavities as the most frequent colonization site in up to 30% of the human population [82,83,84]. Even though *S. aureus* is commensal in humans, it is a frequent cause of a number of clinical manifestations, such as skin and soft tissue infection, pleuropulmonary infection, infective endocarditis, osteomyelitis, and vascular catheter-related infections [85,86]. Treatment of *S. aureus* infection often includes antibiotic therapy and drainage of the infection site. However, managing *S. aureus* infection has become increasingly problematic worldwide due to the emergence of antibiotic resistance in strains such as methicillin-resistant *S. aureus* (MRSA) [87,88,89]. In 2017, the Centers for Disease Control and Prevention (CDC) estimated that more than 323,700 cases in hospitalized patients and more than 10,600 deaths were caused by MRSA in the United States [90].

Previous studies have revealed that *S. aureus* infection is established and maintained through an array of virulence factors such as adhesion proteins, toxins, secretory enzymes, and immune-modulatory factors [91]. In the respiratory system, *S. aureus* infection depends on its ability to breach the airway epithelial barrier by affecting the cell–cell junctions. There is emerging evidence indicating that *S. aureus* damages the airway epithelial barrier by altering the expression of TJ and AJ proteins. For example, it has been shown that conditioned media from *S. aureus* impairs the airway epithelium by disrupting the TJs between primary human nasal epithelial cells (HNECs) cultured at an ALI, in which discontinuous expression of ZO-1 was observed after treatment with conditioned media [26]. A similar impairment in barrier integrity and change in the expression of ZO-1 were noted when purified *S. aureus* V8 protease was added to HNECs in ALI cultures [27]. In a study using ALI cultures of HNECs from patients with severe chronic rhinosinusitis with nasal polyps, *S. aureus* enterotoxin B stimulation was associated with reduced ZO-1 and occludin localization at cell membranes. Similarly, decreased protein levels of claudin-1 and phosphorylation of occludin were also found in patient cells after exposure to *S. aureus* enterotoxin B [21]. This study also examined the effects of *S. aureus* enterotoxin B in mice and found that nasal challenge with *S. aureus* enterotoxin B significantly increased mucosal permeability and decreased mRNA expression of ZO-1 and occludin compared with saline. Furthermore, the authors showed that *S. aureus* enterotoxin B damaged nasal polyp epithelial cell integrity by triggering the Toll-like receptor 2 (TLR2) pathway [21]. A study by Kalsi et al. reported that *S. aureus* infection reduced the expression of ZO-1 and occludin in the monolayer of human airway epithelial cells (H441). Taken together, these in vitro and in vivo data suggest that *S. aureus* impairs epithelial integrity by decreasing the expression of TJ proteins, especially occludin, through secreted viral factors. Interestingly, the antidiabetic drug metformin, which reduces airway glucose permeability and hyperglycemia-induced *S. aureus* load [92], was shown to improve epithelial barrier function by promoting the abundance and assembly of occludin at TJ via an AMPK-PKCζ pathway in human AECs [28].

AJ proteins have also been implicated as targets of *S. aureus* disrupting the epithelial barrier. It has also been reported that *S. aureus* α-toxin (also known as α-hemolysin) binds and activates the metalloproteinase domain-containing protein 10 (ADAM10), resulting in the cleavage of the AJ protein E-cadherin, which disrupts the lung epithelial barrier in mice. In the ADAM10-knockout mice, *S. aureus*-induced E-cadherin proteolysis and barrier disruption were attenuated compared with control animals, suggesting the potential of inhibiting ADAM10 in alleviating epithelial barrier dysfunction caused by *S. aureus* [29]. In a genetic screen sought to identify the host factors that mediate α-toxin cytotoxicity from *S. aureus* [93], several components of the AJ were discovered, including the junctional protein pleckstrin-homology domain-containing protein 7 (PLEKHA7). It was suggested that PLEKHA7 promotes the ADAM10-mediated toxicity of *S. aureus* involving PDZ domain-containing protein 11 (PDZD11) and Tspan33, which is a known factor for α-toxin cytotoxicity [94]. Therefore, strategies to preserve AJ proteins might have therapeutic potential to mitigate *S. aureus* virulence.

Another mechanism that *S. aureus* possibly utilizes to affect intercellular junction and perturb the epithelial barrier is through compromising the actin cytoskeleton. It has been reported that α-toxin and ADAM10 form a complex to disrupt focal adhesion in alveolar epithelial cells, where focal adhesion kinase (FAK) and steroid receptor coactivator (Src) dephosphorylation were observed [30]. This study showed that the α-toxin-ADAM10 complex was assembled in cholesterol/sphingolipid-rich caveolar rafts on the membrane, which likely provided access to deactivate caveolae-associated proteins FAK and Src. It is well known that the activated FAK/Src complex phosphorylates p130Cas and paxillin, which in turn link integrin receptors to Rho family GTPases, actin cytoskeleton, and focal adhesion [95,96]. Therefore, *S. aureus* might disturb barrier integrity by compromising the actin network and disrupting focal adhesions. More recently, *S. aureus* α-toxin-induced actin filament remodeling was revealed in human AECs [31]. In 16HBE14o- human bronchial epithelial (16HBE) cells, *S. aureus* α-toxin led to the hypophosphorylation of cofilin at Ser3 via inhibiting p21-activated protein kinase and LIM kinase activities. Cofilin is an actin-depolymerizing factor, and the dephosphorylation of pSer3-cofilin results in its activation [97]. Therefore, α-toxin caused a loss of actin stress fibers and destabilization of cell shape and cell–cell connection. The authors also noticed that protein kinase A and small GTPases (Rho, Rac, Cdc42) did not seem to be involved in this response. Interestingly, evidence has shown that a dynamic actin cytoskeleton and activation of Src family protein–tyrosine kinases (PTKs) in 293T cells mediate *S. aureus* invasion, and Src PTK-deficient cells are resistant to *S. aureus* infection [98].

Collectively, progress in unraveling mechanisms of *S. aureus* infection in AECs provided potential approaches to combat the epithelial barrier disruption caused by *S. aureus*.

### 2.2. Streptococcus pneumonia

*Streptococcus pneumoniae* (*S. pneumoniae*) are lancet-shaped, Gram-positive bacteria that commonly inhabit the healthy upper respiratory tract in humans [99,100]. As an opportunistic pathogen, *S. pneumoniae* can cause non-invasive and invasive infections, such as otitis media, sinusitis, bacteremia, pneumonia, and meningitis [99,101]. Globally, *S. pneumoniae* is one of the most common causes of community-acquired pneumonia and the leading cause of pneumonia mortality [102]. Typical treatments of *S. pneumoniae* include antibiotic therapies, while the more severe cases might need intensive care unit (ICU)-level care and mechanical ventilation. However, the prevalence of antimicrobial resistance in *S. pneumoniae* has considerably increased since it was first reported in the 1960s [103], posing a growing challenge to the clinical management of *S. pneumoniae* infection [104,105]. A global antimicrobial surveillance program from 1997 to 2016 reported an overall 65.8% penicillin susceptibility in *S. pneumoniae* isolated worldwide in 2015–2016 [106], and the U.S. CDC identified the drug-resistant *S. pneumoniae* as a serious threat with an estimated death toll of 3600 among 900,000 infections in 2014 [90]. Pneumococcal conjugate vaccines (PCVs) against *S. pneumoniae* have significantly reduced invasive pneumococcal disease caused by vaccine-targeted serotypes [107,108]. However, because of the existence of 100 different serotypes [109], the replacement of vaccine serotypes by non-vaccine serotypes has been reported [110,111,112], raising growing concern about a new challenge and urging for a better understanding of pathogenies of *S. pneumoniae* infection.

From their colonization in the upper respiratory tract, *S. pneumoniae* needs to cross the barriers of pulmonary epithelial cells before reaching blood circulation [113]. Previous studies have shown that *S. pneumoniae* impairs the integrity of the epithelial barriers in mice and cultured epithelial monolayers [32]. Separation of the TJs is also observed in human respiratory tissues after *S. pneumoniae* infection [33], and *S. pneumoniae* infection causes dose-dependent AJ disruption in lung epithelial cells in vivo and in vitro [34]. Intercellular junction proteins have been implicated as key targets during the epithelial morphological and functional disruption caused by *S. pneumoniae*. Peter et al. found that pneumococcal infection reduced alveolar ZO-1, occludin, claudin-5, and VE-cadherin when comparing naïve and *S. pneumoniae*-infected human lung tissues using spectral confocal microscopy and Western blot [35]. *S. pneumoniae*-induced cleavage and degradation of E-cadherin [29,36] and reduced claudins 7 and 10 [32] were also reported in lung epithelial cells.

Continuous research endeavors have uncovered possible mechanisms utilized by these bacteria to diminish TJ/AJ components of host cells. In an in vitro model of the pneumococcal invasion process, *S. pneumoniae* was shown to traverse the A549 epithelial monolayers through the facilitation of bacteria-bound plasmin, which disrupted intercellular junctions of epithelial cells by cleaving and degrading E-cadherin [36]. Pneumolysin (PLY), a pore-forming toxin released by *S. pneumoniae*, was also reported to trigger E-cadherin cleavage via activating ADAM10 [29]. Activation of the TLR pathway is another mechanism used by *S. pneumoniae* to get across the epithelium. TLR-dependent decreases of claudins 7 and 10 were observed in a murine model as well as cultured 16HBE monolayers after *S. pneumoniae* infection [32]. This study also revealed that the downregulation of claudins was mediated by an increase in transcriptional repressor Snail1 following the activation of TLR/p38 mitogen-activated protein kinase (MAPK)/TGF-β cascade caused by *S. pneumoniae*. On the other hand, treatment with recombinant interferon-β (IFN-β) inhibited bacterial invasion and transmigration in mouse models [20]. Moreover, in this intranasal infection model, increased mRNA levels of *Cdh1* (E-cadherin), *Tjp1* (ZO-1), *Cldn4* (claudin 4), *Cldn5* (claudin 5), and *Cldn18* (claudin 18) were observed after IFN-β treatment. These data suggest that increased expression levels of TJ and AJ proteins might be of therapeutic interest in combating *S. pneumoniae* invasion. Interestingly, perijunctional F-actin, a cytoskeletal structure connected to TJ and AJ, was severely distorted during the first 3–4 h after *S. pneumoniae* infection prior to the detection of deficits in barrier integrity. As an essential structural foundation for the maintenance of barrier integrity, perijunctional F-actin could be another target to mitigate the structural and functional barrier damage at an early stage of *S. pneumoniae* infection [114].

### 2.3. Pseudomonas aeruginosa

*Pseudomonas aeruginosa* (*P. aeruginosa*) is a Gram-negative, aerobic rod bacterium of the *Pseusomonadaceae* family and a member of the Gammaproteobacteris [115]. *P. aeruginosa* is commonly found in the environment as an inhabitant in soil, water, and plants. *P. aeruginosa* is an emerging opportunistic pathogen that causes recurrent or persistent infections in patients with cystic fibrosis (CF), primary immunodeficiency, diabetes mellitus, severe burn, and those affected by cancer and acquired immunodeficiency syndrome (AIDS) [116,117].

*P. aeruginosa* causes infections in various organs, such as the respiratory system, urinary tract, gastrointestinal tract, skin, soft tissue, bone, joints, and blood [116]. The epidemiological data show that *P. aeruginosa*, particularly antibiotic resistance strains, can cause widespread nosocomial spread [116]. Indeed, nosocomial *P. aeruginosa* is the fourth most commonly isolated pathogen and the most common Gram-negative organism, accounting for 10% of all hospital-acquired infections, especially in ICU patients. In 2017, the CDC estimated 32,600 infections among hospitalized patients and 2700 deaths in the United States [90].

Adhesion to the airway epithelium is a critical step for *P. aeruginosa* infection. In addition to causing infection and inflammation, there are multiple reports of the effect of *P. aeruginosa* on the airway epithelial barrier. One study showed that *P. aeruginosa* infection decreased the transepithelial electric resistance (TEER) and increased the paracellular glucose flux across the Calu-3 airway epithelial cells. Furthermore, *P. aeruginosa* infection decreased the expression of occludin and claudin-1 and induced the cleavage of occludin but had no effect on E-cadherin expression [37]. These effects of *P. aeruginosa* on Calu-3 TJ were thought to be through the induction of hyperglycemia since pre-treatment with the antidiabetic drug metformin decreased the *P. aeruginosa*-induced barrier disruption and inhibited bacterial growth [37]. This is consistent with the data that elevated bronchial glucose is associated with an increased risk of respiratory infection in patients admitted to intensive care patients [118,119]. In CF patients who are at risk of *P. aeruginosa* colonization and infection, the elevated blood glucose concentrations in airway surface liquid (ASL) associated with cystic fibrosis-related diabetes (CFRD) has been shown to increase bacterial growth such as *P. aeruginosa* and *S. aureus*. This phenomenon was thought to be secondary to the impact of hyperglycemia on the cystic fibrosis transmembrane conductance regulator (CFTR), which has an integral role in airway barrier function [120]. Impaired CFTR could also lead to viscous ASL and enhance infection susceptibility [37]. The lack of proper CFTR function, viscous ASL, and impaired epithelial barrier prompts airway colonization with multiple bacteria, including *P. aeruginosa*. Another study compared the impact of *P. aeruginosa* and *E. coli* infection on airway barrier structure and function using in vitro and in vivo models. The in vitro studies showed a decreased TEER, damaging ZO-1 structure, and increased permeability to bacterial infections through the epithelial layer. At the same time, *E. coli* infection did not affect the epithelial barrier. Similarly, the in vivo studies showed that C57BL/6 mice infected with *P. aeruginosa* exhibited an increased permeability to fluorescein isothiocyanate (FITC)-labeled polyplexes into the lung parenchyma, indicating disruption of airway epithelium, while *E. coli* infection did not affect airway barrier function [38]. Nomura et al. applied *P. aeruginosa* elastase (PE) on HNECs cultures. They observed a transient decrease in TEER associated with a reduction in the expression of claudin-1 and -4, occludin, and tricellulin. Interestingly, PE exposure did not affect ZO-1, ZO-2, E-cadherin, or β-catenin. The effect of PE on TJs was attenuated by chemical inhibitors of PKC, MAPK, phosphoinositide 3-kinase (PI3K), p38 MAPK, JNK, COX-1 and -2, and NF-κB pathways. There was evidence of decreased PAR-2; consequently, the PE-induced TJ disruption was attenuated by a PAR-2 agonist, while the PAR-2 knockdown, even in the absence of PE, downregulated the TJs [39]. Likewise, Clark et al. found that exposure of Calu3 cells to PE induced PKC signaling, disrupted occludin and ZO-1 proteins, and caused actin cytoskeletal reorganization [40].

The endotoxins released by *P. aeruginosa,* such as rhamnolipids, could also disrupt TJ structure and decrease barrier function, especially in patients with CF who are chronically colonized with these bacteria [41]. This evidence was supported by observations that adding merely the purified rhamnolipids could trigger a reduction in TEER in a dose and time-dependent manner and disrupt the TJ structure [41]. Moreover, to establish their interaction with the host and deliver cytotoxins directly into eukaryotic cells to mediate the pathogenesis, Gram-negative bacterial secrete multiple effector proteins via their Type III Secretion System (T3SS) apparatus. Four effectors, protein Exoenzyme (Exo) S, ExoT, ExoU, and ExoY, are used by *P. aeruginosa* strains to mediate the pathogenesis [121]. For example, the Exo S and ExoT disrupted the actin cytoskeleton of host AEC and induced barrier disruption by impairing cell-to-cell adhesion, and ExoY disrupted the barrier integrity without cytotoxicity, while another effector protein, ExoU, exerted rapid necrotic cytotoxicity [121,122,123,124,125,126]. Similarly, another study showed that type III toxins, especially ExoS-induced permeability in 16HBE cells, altered the distribution of ZO-1 and occludin [42].

### 2.4. Burkholderia

*Burkholderia*, a genus of *Pseudomonadota*, contains more than 80 Gram-negative coccobacilli species, which are ubiquitous within the environment and pathogenic to plants, insects, animals, and humans [127,128]. *Burkholderia cepacia* complex (BCC) is a group of at least 24 species. One of them is *B. cepacia*, which is an opportunistic human pathogen that most often causes pneumonia and fatal infections in individuals with CF, chronic lung disease, sickle cell, and immunodeficiency. These bacteria can also cause severe nosocomial infections, bacteremia, and sepsis [129]. In 20% of patients, the bacteria can trigger cepacia syndrome, which is characterized by fatal necrotizing pneumonia with bacteremia [128,130]. Antibiotic therapy for these bacteria is challenging due to the intrinsic antimicrobial resistance and acquired resistance to antimicrobial agents [131]. Additionally, some CF patients can be infected with *B. cepacia* and *P. aeruginosa* together, which causes more severe infection.

It has been shown that *B. cepacia* can pass through airway epithelium; however, the involved mechanisms still need to be better understood. Mounting evidence suggests that *B. cenocepacia* can cross the respiratory epithelium by disrupting TJs. Kim et al. utilized polarized 16HBE cells in vitro and found that *B. cenocepacia* infection decreased TEER and increased the flux of FITC-labeled bovine serum albumin (BSA) across the cell monolayer [43]. They did not observe significant cytotoxicity or cell death in infected cells. Confocal fluorescence microscopy observed a substantial decrease in occludin expression, while ZO-1 remained intact. Furthermore, they saw the dephosphorylation of occludin and concluded that *B. cenocepacia*-induced dephosphorylation, and occludin dissociation could facilitate the migration of bacteria through the respiratory epithelium, leading to bacteremia [43]. Sajjan et al. used lung explants from CF recipients who underwent lung transplants with and without *B. cepacia* and autopsy sections of CF patients with *B. cepacia* infection [44]. They showed evidence that *B. cepacia* altered the epithelial cell barrier by increasing the permeability, which allows bacteria to migrate through the polarized respiratory epithelium to the lung interstitium and lumen of blood capillaries and cause blood infection and sepsis [44]. Another study used three strains of *B. cepacia* complex known to cause cepacia syndrome to study the mechanisms of bacteremia caused by these organisms. Using well-differentiated human AEC cultures, they showed that these strains could invade the epithelial barrier via different invasion pathways. For example, the *B. cepacia* BC-7 formed a biofilm near the apical cell surface, attached to the cell surface, disrupted the superficial epithelial cell layer, and dislocated the actin cytoskeleton. In contrast, the HI2258 strain (genomovar IV) did not form biofilms and passed through the between epithelial cells without disrupting the integrity of the epithelium, suggesting a paracytosis route. However, strain J-1, with both biofilm-dependent and independent ability, invaded the tissue by both formation of biofilm as well as inducing both paracytosis and cell destruction [45].

In a similar study, Duff et al. used three lung epithelial cells to examine mechanisms by which BCC invades the epithelial barrier [46]. Four species of BCC were used in this study, and all caused a decrease in TEER. However, the investigators showed that some of these species could easily traverse the cell monolayer, while others were slower to translocate, which suggests differences in their potential to invade the epithelial barrier. Confocal microscopy and Western blot analysis showed a decreased expression of ZO-1 and E-cadherin in Calu-3 cells [46], which contrasts with undisrupted ZO-1 observed in 16HBE cells by Kim et al. [43], and it might suggest a cell-specific difference or various study time points. Of note, Duff et al. showed data for confocal studies and Western blot analysis of ZO-1 using Calu-3 cells, not 16HBE cells.

Blume et al. infected 16HBE and primary human bronchial epithelial cells (HBECs) with *Burkholderia thailandensis* (*B. thailandensis*) and found a dose- and time-dependent decrease in TEER. Additionally, they observed an increase in the bacterial traverse from the apical to the basolateral compartment, which was associated with a dose-dependent increase in TNF-α cytokine. The effect of *B. thailandensis* on TEER was marginally modified by pre-treatment with a TNF-α neutralizing agent (etanercept). In contrast, pre-treatment with a corticosteroid (fluticasone propionate) significantly prevented the decrease in TEER. These data suggest that TNF-α neutralizing agents or steroids might be a therapeutic option to maintain airway barrier functions and reduce inflammation [47].

One study compared the impact of two environmental strains with a clinical strain of *B. cenocepacia* on non-CF bronchial epithelial cells, 16HBE, and CF cell line, CFBE41o−. They observed a drop in TEER in cells exposed to environmental strains similar to the clinical strain. Interestingly, in this study, the extent of decrease in TEER was comparable between CF and non-CF cell lines. In contrast, the ZO-1 disruption was more pronounced in cells infected with clinical strain than environmental strains. The ZO-1 distribution seemed to be disrupted at baseline in the CF cell line, with more significant disruption in infected cells [48]. Another study infected the CFBE41o- cell line with *B. contaminans* strains and saw a time-dependent decrease in TEER and an increase in FITC-labeled BSA flux across the cell monolayers. They also observed reductions in ZO-1, occludin, and claudin-1 expressions [49].

Taken together, these studies show evidence that infection with *Burkholderia* species decreases airway epithelial barrier integrity and increases proinflammatory cytokines that contribute to bacteremia, infection dissemination, and airway inflammation.

### 2.5. Haemophilus Influenzae

*Haemophilus influenzae* (*H. influenzae*) is a Gram-negative, coccobacillus opportunistic anaerobic pathogenic bacterium of the family *Pasteurellaceae* [132]. There are two major strains of *H. influenzae* based on the presence or absence of a distinct capsular polysaccharide antigen [132]. The encapsulated strains are classified based on their specific capsular antigens into six distinct groups designated as serotypes a–f, with type b being the most common strain. The unencapsulated strains are termed non-typeable *Haemophilus influenzae* (NTHi) since they do not have a capsular serotype [133]. The encapsulated strains usually cause more invasive infections. While the unencapsulated strains are less invasive, they can trigger inflammatory responses in humans, leading to many symptoms. In the pre-vaccine era, *H. influenzae type b* (Hib) was the leading cause of severe bacterial infections such as pneumonia and meningitis among children. However, since the arrival of an effective Hib conjugate vaccine in 1988, the NTHi strains have been the most common cause of bacterial respiratory tract infection in young children as well as exacerbations of chronic obstructive pulmonary disease (COPD) and bronchiectasis in adults [133]. In addition, the NTHi can often cause more invasive diseases such as meningitis and sepsis. Currently, there are no vaccines to protect against NTHi, since there are many strains of NTHi with significant genetic heterogenicity; thus, the antibody against one particular strain does not protect against other strains [134].

The pathogeneses of NTHi are not well understood, but it is known that these bacteria possess several adhesive factors which promote colonization, leading to infection. Previous studies have shown that NTHi infection of the lower airways could cause chronic airway inflammation and even cause susceptibility to other infections [135,136]. Alveolar epithelium is composed of a mixed monolayer of type I and type II alveolar epithelial cells. Kaufhold et al. used an in vitro model of alveolar epithelial cells type II and A549 cells to study the role of NTHi on airway barrier function [50]. They showed that infection with NTHi decreased E-cadherin mRNA expression and protein levels. Additionally, immunofluorescence staining 24 h post-NTHi infection disrupted E-cadherin and ZO-1 structure. This was thought to be secondary to the upregulation of fibroblast growth factor 2 (FGF2) and activation of the mTOR pathway, since a pharmacological inhibitor of mTOR, rapamycin, prevented the reduction in E-cadherin expression [50].

It has been shown that NTHi activates the NF-κB signaling pathways by either NF-κB translocation-dependent or -independent pathways [137]. NF-κB is a known transcription factor regulating cell responses to inflammation. Studies conducted by Ward et al. showed that inhibiting IκB kinase, an enzyme complex that forms part of the NF-κB signaling pathway, triggered a dose-dependent decrease in TEER in the absence of cell death [138]. In a comprehensive study, the authors used primary rat type II alveolar epithelial cells to generate a model of type I cell monolayers and analyzed the effect of IκB kinase inhibitors on several TJ proteins. For example, immunofluorescence microscopy revealed the disruption of claudin-18, ZO-1, and ZO-2 as well as cortical actin cytoskeletal rearrangements and alteration of β-catenin protein [138]. Studying various TJ proteins, the investigators found an increase in claudin-4 and claudin-5 and a decrease in claudin-18 with altered TJ morphology. An increase in claudin-4 was thought to be a proinflammatory response. In contrast, a claudin-5 increase is associated with an increase in the paracellular leak in alveolar epithelial cells, and a decrease in claudin-18 correlates with an impaired barrier function [138,139,140,141,142]. There is evidence that the AJ protein E-cadherin expression is reduced in COPD patients [143]. A recent study by Glockner et al. analyzed the impact of NTHi on the re-differentiation of epithelial cells and airway remodeling. Using primary human HBECs grown at ALI infected with NTHi, they studied the E-cadherin as an epithelial marker [144]. An upregulation of mesenchymal marker vimentin in HBECs was observed, which suggests airway remodeling. They also found that NTHi infection did not change the expression or stability of E-cadherin. However, this study did not directly analyze the role of E-cadherin changes in the epithelial barrier.

Multiple groups have revealed the importance of airway epithelial barrier and differentiation in asthma [2,145,146]. Asthma was previously recognized as a T helper 2 (Th2) high eosinophilic inflammatory disease responding to glucocorticosteroids. However, a severe neutrophilic phenotype, not responsive to current therapeutic modalities, has been established in a large subgroup of asthmatic individuals [147]. A recent paper reviewed the role of persistent bacterial infection in asthma [148]. It summarized evidence that NTHi can adhere to respiratory tract epithelium, upregulate neutrophil chemokines and proinflammatory cytokines, and induce a persistent infection. Clinical trials have observed the benefits of using antimicrobials such as azithromycin in reducing asthma exacerbation, likely by decreasing the *H. influenzae* load and reducing inflammatory cytokines [149,150]. However, due to frequent side effects and the risk of antimicrobial resistance development, azithromycin has been considered only in a selected group of asthmatics [151]. This evidence, combined with the impact of NTHi on reducing the expression of TJ proteins, suggests that NTHi infection plays a critical role in inducing chronic neutrophilic asthma by impairing the airway epithelial barrier. Therefore, investigating and targeting chronic NTHi infection and airway dysfunction in severe persistent neutrophilic asthma is critical.

## 3. Viruses

### 3.1. Respiratory Syncytial Virus

Human respiratory syncytial virus (RSV) is an enveloped, negative-strand RNA virus of the genus *Orthopneumovirus*, family *Pneumoviridae* [152] that transmits through aerosolized droplets [153,154]. RSV infection affects the lungs and respiratory tract and often causes mild, cold-like symptoms in healthy populations. However, RSV can lead to severe infections and cause pneumonia or bronchiolitis in young children and other vulnerable groups, such as premature infants, older adults, and individuals with underlying immune, cardiac or pulmonary conditions [155]. As a common virus, RSV infection poses a significant burden worldwide. RSV is the leading cause of acute lower respiratory tract infection in infants and young children globally and a major cause of hospitalization and mortality in the elderly and immunocompromised population [156,157,158,159,160]. RSV infection has also been associated with persistent wheezing and asthma [161,162,163,164]. There are multiple other factors contributing to RSV pathogenesis and disease severity [165]. For example, the severity of RSV-associated symptoms is influenced by host factors such as Vitamin D level and genetic predispositions [166]. Environmental factors such as exposure to air pollution and tobacco smoke have also been recognized as key players with the potential to increase the risk of RSV infection and exacerbate clinical manifestations related to RSV [167,168,169,170,171].

The current management for RSV is primarily supportive because there are no effective antiviral therapies or vaccines against RSV yet [172,173]. So far, aerosolized ribavirin and palivizumab are the only two antiviral drugs approved by the U.S. FDA for treating or preventing severe respiratory tract infections caused by RSV [174]. Ribavirin is the only licensed drug to treat RSV infection; however, the clinical application of ribavirin is highly limited by its unproven efficacy, non-specificity, and potential toxicity [155,175,176]. Palivizumab is the only product available for preventing RSV infection despite rapid expansion in the development of RSV vaccines [157,173]. This monoclonal antibody (mAb) has been shown to reduce hospitalization, recurrent wheezing, and perhaps nonatopic asthma caused by RSV infection in certain high-risk infants [164,177]. However, the application of palivizumab is restricted to small high-risk populations, leaving the majority of infants unprotected [178,179]. These unmet clinical needs highlight the significance and urgency to understand the pathological mechanism of RSV infection.

The primary targets of RSV are ciliated airway epithelial cells and type I alveolar cells [180,181,182]. Transcription and replication start rapidly after RSV enters the epithelial cells, which is followed by the assembly and release of viral particles from host cells. Over time, the virus will breach the epithelial barrier and cause widespread inflammation and pulmonary damage if not cleared in time [1,15,183]. Using in vitro cultures of primary HBECs and human bronchial cell lines such as 16HBE, multiple groups, including us, showed that RSV infection decreased the TEER and increased the permeability of the monolayers without obvious cytopathology [15,18,51,52,53]. In rodent models, intranasal RSV infection increased the permeability of the airway epithelial barrier in C57BL/6 mice [15] and BALB/c mice [54].

The evident disassembly of AJC was observed during RSV infection and caused the “leaky” barrier in these models. Moreover, emerging evidence indicates that changes in the molecular components of epithelial TJs and AJs occur during RSV infection. Increased claudin-2 (a pore-forming claudin whose overexpression increases paracellular permeability), as well as decreased protein levels of ZO-1, occludin, and claudin-1, were described in lung tissues from RSV-infected C57BL/6 mice, while the levels of claudin-18, β-catenin, and E-cadherin were not significantly altered [15]. Another murine model of RSV infection exhibited decreased mRNA of claudin-1 and occludin in lung samples compared with control animals [54]. Interestingly, unlike in vivo observations, RSV infection did not impose significant impacts on the protein levels of TJ and AJ proteins in bronchial epithelial monolayers, including ZO-1, occludin, E-cadherin, and β-catenin [18]. Others examined RSV infection of cultured nasal epithelial monolayers and reported an increase in claudin-4 but no change in the expression of other AJC proteins [55]. This discrepancy might be caused by differences between in vitro and in vivo models of RSV infection, such as the inflammatory response to viral infection in vivo. For example, IL-4, IL-13, and IFN-γ are known to disrupt TJ integrity and decrease the expression of different junctional proteins [184,185,186]. Furthermore, these studies suggest alternative mechanisms for RSV-induced barrier dysfunction besides dysregulated expressions of TJ and AJ proteins.

Recently discovered evidence indicates that AJC disorganization by RSV could result from TJ/AJ structural rearrangements caused by perijunctional F-actin cytoskeletal remodeling. TJ and AJ are anchored to the actin cytoskeleton network, especially the perijunctional F-actin bundles, and it is well acknowledged that this connection is a critical regulator of AJC structure and epithelial barrier function [7,9,187]. For example, emerging evidence suggests that RSV induces the depolymerization of perijunctional F-actin, and pharmacologically stabilizing F-actin filaments modify RSV-induced barrier dysfunction [53]. Previously, our group showed evidence of protein kinase D (PKD) activation during RSV infection in 16HBE cells, while PKD antagonists protected cells from barrier disruption and AJC disassembly caused by RSV [18]. PKD has been implicated in cell adhesion, cell motility, and invasion via regulating actin remodeling [188,189,190]. Others have shown that PKD controls actin polymerization via phosphorylation of cortactin [191], which is an actin-binding protein and a key player in regulating F-actin dynamics and promoting actin assembly [192,193,194,195]. Indeed, cortactin phosphorylation was observed in association with PKD activation during RSV infection [18]. Therefore, the PKD pathway could be an upstream regulator of F-actin cytoskeletal remodeling whose activation disrupts AJC and airway epithelial barrier during RSV infection. RSV has been revealed to destabilize the perijunctional F-actin network and lead to a disrupted epithelial barrier by inhibiting cyclic adenosine monophosphate (cAMP)/Rap1/Rac1 signaling [53]. This study found that activating Rap1 by cAMP analog 8-pCPT-2-O-Me-cAMP-AM mitigated RSV-induced epithelial barrier disruption, which was in agreement with findings that elevating cAMP levels by adenylyl cyclase activator forskolin or cAMP analogs 8-Bromo-cAMP could protect airway barrier function and reduce RSV replication in cultured epithelial cells [51]. Considering the crucial role of cAMP/Rap1/Rac1 signaling in controlling the dynamics of the F-actin cytoskeleton, the cAMP pathway could be an essential mechanism involved in the RSV-induced reorganization of the F-actin network to interrupt the TJ and AJ. Collectively, these studies point out that RSV-induced AJC dysfunction could be regulated by the cAMP pathway and PKD pathway via perijunctional F-actin cytoskeletal remodeling. Moreover, cortactin expression decreased in RSV-infected cell monolayers and mouse airways and was implicated as the cause for the reduced activity of Rap1 [53], implying possible cross-talks between these two pathways. Accordingly, these studies provide insights into the therapeutic potentials of forskolin, cAMP analogs, and PKD inhibitors in RSV management through junction proteins, especially forskolin, which showed protective effects on the barrier integrity even when added 24 h after RSV inoculation [51].

Other signaling pathways are also implicated to be responsible for RSV-induced airway epithelial barrier disruption. For instance, the cleavage of E-cadherin and accumulation of soluble extracellular fragments of E-cadherin (referred to as soluble E-cadherin) were detected both in vivo and in vitro after RSV infection, and adding recombinant soluble E-cadherin to cell culture caused TJ disassembly in primary HBECs monolayers [15]. It is not clear about the protease(s) responsible for this cleavage, but it would be interesting to explore the impacts of limiting the soluble E-cadherin on RSV-induced barrier deficits. Others report that RSV infection increased bronchial epithelial monolayer permeability in vitro via inducing vascular endothelial growth factor (VEGF) [56] and activating the p38 MAPK pathway [52]. However, whether and how TJs and AJs are involved in these pathways warrant further investigations. Other cell types play an important role in RSV virulence and could be targeted for disease management. RSV induces the production of proinflammatory mediators in AECs and recruits immune cells, including neutrophils [196]. Indeed, significant neutrophil infiltration has been observed in the lungs of infants with severe RSV-induced bronchiolitis [197]. In a co-culture assay system of neutrophils and RSV-infected A549, Deng et al. found that trans-epithelial migration of neutrophils deteriorated RSV-induced epithelial barrier disruption [198].

### 3.2. Human Rhinovirus

First identified in the 1950s, human rhinovirus (HRV) is a member of the *Enterovirus* genus in the *Picornaviridae* family [199,200]. This non-enveloped virus contains a positive-sense, single-stranded RNA [200,201]. As a ubiquitous virus, HRV transmits through direct contact or aerosol [202] and causes infections year-round [203,204,205]. HRVs are the most common cause of upper respiratory tract infections [206,207]. Although HRV is not typically considered to cause high mortality, it has been implicated in lower respiratory diseases including pneumonia, bronchiolitis, asthma, and COPD [208,209,210,211,212,213], indicating its potential to cause acute or chronic respiratory diseases. Currently, HRV infection management is mainly supportive care, because there is no affordable and effective therapy against RVs [214]. Numerous antiviral reagents against HRVs have been tested but showed limited efficiency or severe side effects [207]. At the same time, the development of an HRV vaccine is precluded by the number of different HRV serotypes [214].

The primary targets of HRV infection and replication in humans are AECs in both upper and lower airways [215,216,217]. HRV is reported to infect both ciliated and non-ciliated epithelial cells [218]. HRV has been described to impair barrier integrity by disrupting TJs and AJs. Previous studies in polarized primary human AECs and 16HBE cells showed that HRV infection reduced TEER and increased paracellular permeability without causing detectable cytopathology [57,58,59,60]. However, others observed a cytopathic effect of HRV infection in association with comprised barrier integrity [61,219]. Despite the inconsistent observations in cell toxicity, ample evidence indicates that HRV infection-dependent impairment of the airway epithelial barrier correlates with disruption of the expression and localization of TJ/AJ proteins. Both in vitro and in vivo studies have shown that HRV caused the dissociation of ZO-1 from TJ and consequently led to the loss of epithelial barrier integrity of nasal epithelial cells [58]. In primary HNECs grown at ALI, Yeo and Jang observed decreased mRNA and protein levels of ZO-1, occludin, claudin-1, and E-cadherin after HRV infection [62]. Similarly, another study reported HRV-induced reduction in ZO-1, occludin, claudin, and E-cadherin proteins in primary HNECs [63]. Consistently, the reduced expression of Crumbs 3, ZO-1, occludin, claudins 1 and 4, and E-cadherin was observed after HRV infection in an in vitro model of injured/regenerating airway epithelium [57]. Looi et al. also observed a reduced protein expression of TJ proteins ZO-1, occludin, and claudin-1 following HRV infection in both human airway epithelial cell line NuLi-1 and primary AECs obtained from patients [61]. Interestingly, HRV infection caused a transient dissociation of TJ in primary AECs from non-asthmatic children but a significant reduction in ZO-1 and occludin expression in cells from asthmatic children [64], correlating to the relationship between HRV infection and exacerbated barrier dysfunction in children with asthma. In HRV-infected human precision-cut lung slices (PCLS), the gene expression of claudin-8 was significantly downregulated compared with that in control tissues [65]. This series of evidence implies a critical role of HRV-induced disruption of AJC in the development and progress of HRV-related respiratory diseases.

Understanding the molecular mechanisms of HRV-induced AJC disassembly and downregulation of TJ/AJ protein expressions is important to the discovery of new therapeutic targets. Oxidative stress could lead to the impairment of airway epithelial barrier function [167,220,221], and HRV has been shown to induce oxidative stress in AECs [222]. Previous studies revealed that HRV disrupted the barrier function of polarized 16HBE monolayer by stimulating reactive oxygen species (ROS) generation, possibly through dsRNA generated during HRV replication [66,67]. NADPH oxidase (NOX) mediates the production of ROS and contributes to respiratory viruses-induced barrier disruption through inflammatory pathways [223,224]. Comstock et al. reported that HRV infection induced ROS generation, and inhibiting NOX or NOX1 blocked the disruptive effects of HRV on barrier integrity and the dissociation of ZO-1 and occludin from AJC [66]. In addition to the NOX-1-dependent ROS mechanism, Unger et al. provided evidence that mitochondrial ROS generation stimulated by HRV also contributes to the disruption of airway epithelial barrier function [67]. Consistently, Kim et al. reported that HRV decreased TJ/AJ proteins in primary HNECs by inducing NOX-derived ROS production. They showed that HRV infection induced NOX-derived ROS production, and inhibiting NOX prevented an HRV-induced decrease in TJ/AJ proteins. Furthermore, the ROS-dependent inhibition of phosphatases is closely linked to reduced TJ/AJ proteins caused by HRV [63]. Taken together, diminishing ROS might show protective effects on epithelial cells against HRV and serve as a therapeutic strategy. Indeed, the role of anti-oxidants in HRV-induced barrier disruption has been investigated. Pretreatment with Ginsenoside Re, a ginsenoside found in Panax ginseng with anti-oxidant actions [225], decreased HRV-induced disruption of TJ/AJ proteins through inhibiting ROS-mediated phosphatases inactivation in HNECs [68]. Wogonin is a flavonoid-like chemical with anti-oxidant and anti-inflammatory properties [226]. A recent study revealed that wogonin prevented increases in intracellular ROS following HRV infection and alleviated the decreases of AJ/TJ proteins by suppressing the phosphorylation of ROS-mediated pathways Akt/NF-κB and ERK1/2 in HNECs [69]. In addition to oxidative stress, other mechanisms have also been implicated in mediating HRV-induced AJC changes. The upregulated expression of Snail, a transcriptional repressor of TJ and AJ proteins, was reported after HRV infection of injured/regenerating airway epithelium [57]. Whether Snail contributes to HRV-induced TJ/AJ change in regular AECs still warrants further investigation. The qPCR arrays and pathway analysis were performed on HRV-infected cells to evaluate associations between human airway epithelial TJ proteins and antiviral response. Network analysis suggested that HRV infection decreased TJ protein and increased epithelial permeability potentially via antiviral responses of IL-15 [61]; however, further experiments, including the addition of neutralizing antibodies, are needed to provide additional evidence for therapeutic potentials. Emerging omics approaches have also provided some new perspectives. For example, proteomic and metabolomic analysis of HBECs at ALI characterized altered metabolic pathways during HRV infection. This investigation revealed PGC-1α, a key mitochondrial biogenesis transcriptional coactivator, as a novel antiviral target, as investigators found that several HRV strains reduced PGC-1α expression, and promoted PGC-1α expression could restore barrier function during HRV infection [60].

### 3.3. Influenza Viruses

Influenza viruses are enveloped, negative-sense RNA viruses with a segmented genome [200,227]. As the major representative of the *Orthomyxoviridae* family, influenza viruses are grouped into four genera: *influenzavirus A*, *influenzavirus B*, *influenzavirus C*, and *influenzavirus D*. [200]. Three types of influenza viruses (A, B, and C) infect humans. Type A and B viruses cause significant morbidity and mortality annually worldwide and are referred to as seasonal influenza viruses [228]. Influenza A (IAV) and B viruses are generally believed to be transmitted at a short range (1–2 m) from person to person through large droplets and aerosols [229]. The clinical manifestations of seasonal influenza cover a wide spectrum, from asymptomatic infection, and uncomplicated upper-respiratory-tract symptoms to complications that can result in severe diseases such as lethal pneumonia, acute respiratory distress syndrome (ARDS), and secondary bacterial infection of the lower respiratory tract [227,228,230,231]. IAVs are the only influenza viruses known to cause flu pandemics and have been the focus of investigations. The World Health Organization (WHO) estimated that annual influenza epidemics cause 3–5 million cases of severe illnesses and up to 300,000–650,000 deaths worldwide [174,232]. In the United States alone, the 2019–2020 influenza season was estimated to cause 16 million medical visits and 25,000 deaths [233]. The segmented genome of influenza viruses enables antigenic shift through reassortment, which mediates cross-species transmission and is associated with influenza A pandemics. Influenza virus also evolves through accumulating mutations, which is a process known as antigenic drift caused by RNA polymerase infidelity [227,234]. Due to such genetic diversity and rapid viral evolution, influenza infections remain a severe threat to public health despite effective antiviral treatments and vaccines being available.

Human influenza viruses target epithelial cells of the upper and lower respiratory tract for infection and replication [227,235]. IAV infection has been well-documented to damage epithelial junctions and impair the barrier function of airway epithelium [113]. In well-differentiated normal human bronchial epithelial (NHBE) cells, infection with the 2009 pandemic IAV/H1N1pdm09 strain led to a decline in barrier function, which was marked by a decrease in TEER and an increase in permeability [17]. This study also showed disrupted staining of ZO-1 and F-actin cytoskeleton as well as cell death after IAV infection. In another study, primary human alveolar type II cells were infected by two strains of H1N1pdm09 viruses, and researchers observed a similar decrease in TEER and an increase in barrier permeability [70]. In an in vitro co-culture model of human alveolar epithelial cells and endothelial cells, Short et al. showed that IAV infection damaged the barrier integrity independently of endothelial cells [16]. Although IAV infection of epithelial cells can induce cell death [236,237], this study detected no apparent cell death after IAV infection, indicating that the barrier disruption was not due to cell death. The investigators also reported that significantly decreased TEER and increased barrier permeability caused by IAV infection were associated with the disruption of TJs amongst epithelial cells, specifically with loss of TJ protein claudin-4 [16]. Interestingly, they found that the barrier damage caused by IAV was independent of the activation of proinflammatory cytokines, indicating the direct effect of IAV in reducing claudin-4 expression through unknown mechanisms.

Despite these observations, the molecular mechanisms of IAV-induced damage to the epithelial barrier are far from well understood. Using cultured A549 cells and an in vivo mouse model, Ruan et al. revealed that IAV H1N1 infection activated MAPK and PI3K signaling, which induced the activation of Gli1, a transcription factor in the sonic hedgehog signaling pathway, and subsequent Snail expression [71]. Increased Snail decreased the expression of AJ protein E-cadherin and TJ proteins ZO-1 and occludin and increased paracellular permeability. Intraperitoneal injection of GANT61, a Gli1 Inhibitor, blocked IAV-Induced expression of Gli1 and Snail and restored expression of E-cadherin and occludin in vivo. GANT61 treatment also suppressed the pathological changes in IAV H1N1-infected lungs [71]. Whether Gli1 can be targeted for antiviral therapy for the H1N1 virus is still unknown and warrants further investigation. The H5N1 virus is a newly emerged strain of IAV from animal reservoirs and raised great concerns about a severe pandemic. A recent study reported that H5N1 viruses reduced the expression of ZO-1, occludin, claudin-1, and E-cadherin in A549 cells, NL20 cells (a human noncancerous alveolar epithelial cell line), and the lungs of H5N1 virus-infected mice [72]. The authors showed evidence that H5N1 virus infection impaired the alveolar epithelial barrier by accelerating the turnover of several AJC proteins. This study suggested a mechanism that H5N1 viruses activated TAK1 and its downstream MAPK, p38, and ERK, leading to the increased expression of E3 ubiquitin ligase Itch and ubiquitination of occludin, claudin-1, and E-cadherin proteins. Furthermore, inhibiting the TAK1–Itch pathway restored the epithelial barrier structure and function in vitro and in vivo [72]. However, whether the TAK1–Itch pathway can be a valid antiviral targeted for the H5N1 virus is uncertain.

### 3.4. Human Parainfluenza Virus

Human parainfluenza viruses (HPIVs) are enveloped, negative-sense, single-stranded RNA viruses and members of the *Paramyxoviridae* family. HPIVs are classified into four distinct serotypes: HPIV-1, HPIV-2, HPIV-3, and HPIV-4 [238,239]. In addition, another parainfluenza virus 5 (PIV5) infects animals such as dogs, pigs, cats, and sometimes humans [240]. HPIVs cause illnesses such as pharyngitis, croup, tracheobronchitis, bronchiolitis, croup, pneumonia, and respiratory distress syndrome. HPIVs transmission occurs through direct person-to-person contact or large droplets or indirect contact via contaminated surfaces [241].

HPIV infects the ciliated epithelial cells and causes inflammation, which contributes to the disease pathogenesis. A couple of studies have explored the impact of tight HPIV infectivity on AEC barrier integrity. One study showed that the exposure of AECs to HPIV3 and PIV5 triggered a reduction in TEER and increased permeability. The study did not analyze the AJC structure or involved pathways [73]. Another study infected human alveolar epithelial cells (A549 cells) with HPIV2 and found an induction of claudin-1 mRNA expression by HPIV2. However, there were no changes in the expression of other TJs, such as ZO-1, ZO-2, occludin, claudin-3, claudin-4, or claudin-7 [74]. The claudin-1 upregulation was thought to be partly induced by a functional HPIV2 V protein, which impacts viral growth, modulates TJ molecules, and facilitates efficient virus propagation. They used a rPIV2 V_C193/197A_, which expressed a mutant V protein and observed an enhancement of CLDN1 mRNA upregulation compared to the wild type. To investigate the importance of claudin-1, they used A549 cell lines that stably over-express CLDN1 and found a decrease in HPIV2 growth and protein expression. Furthermore, in Madin–Darby canine kidney (MDCK) cells, CLDN1 knockout provoked a significant increase in HPIV2 growth and protein expression. Similarly, HPIV2 cell-to-cell spread deceased in A549, overexpressing the CLDN1 and increased in MDCK CLDN1-KO cells, emphasizing the role of claudin-1 [74]. At present, there are no licensed vaccines or antiviral drugs for HPIV infection. Therefore, gaining knowledge of the molecular mechanisms, particularly the role of claudin-1, can aid in developing treatment options to dampen HPIVs growth and spread.

### 3.5. Coronaviruses

Coronaviruses are enveloped, single-stranded, positive-sense RNA viruses of the *Coronaviridae* family [200]. Seven human coronaviruses (HCoV) have been characterized to date. HCoV-NL63, HCoV-229E, HCoV-OC43, and HCoV-HKU1 commonly cause mild upper respiratory illnesses. In contrast, severe acute respiratory syndrome coronavirus (SARS-CoV), Middle East respiratory syndrome coronavirus (MERS-CoV), and severe acute respiratory syndrome coronavirus-2 (SARS-CoV-2) are highly pathogenic and have resulted in outbreaks with high mortality rates [242,243].

SARS-CoV was first reported in southern China in February 2003 [244,245,246] and spread to over 30 countries, leading to over 8000 confirmed cases with a mortality rate of ~9.6% by the end of the epidemic in June 2003 [247,248]. MERS-CoV emerged in Saudi Arabia in 2012 [249,250] and has since caused over 2000 confirmed cases with a mortality rate of ~35% [243,248,251]. SARS-CoV-2 was first identified in China in December 2019 [252] and has caused a massive global pandemic of coronavirus disease 2019 (COVID-19), infecting more than 634 million people and resulting in more than 6.5 million deaths worldwide [253]. The human-to-human transmissibility of MERS-CoV is low [254], whereas SARS-CoV and SARS-CoV-2 readily transmit via the respiratory tract through close contact with infectious droplets and aerosols [255,256]. The clinical manifestations of SARS, MERS, and COVID-19 share many similarities; acute respiratory tract infections are the most common symptoms and could lead to rapid respiratory failure and mortality [243,257]. To date, there is no vaccine or specific antiviral drug available for either SARS-CoV or MERS-CoV [174,248]. The rapid scientific response to COVID-19 enabled the development of highly effective vaccines and drugs at an unparalleled speed [258]. Currently, COVID-19 management includes antivirals such as Molnupiravir and Paxlovid in early stages and immunotherapeutic agents such as IFNs and corticosteroids in more advanced stages [259,260,261]. Despite the significant progress in combating SARS-CoV-2, the emergence of its mutant variants has become a concerning issue and threat to public health.

SARS-CoV utilizes the angiotensin-converting enzyme 2 (ACE2) as a receptor and primarily infects ciliated bronchial epithelial cells and type II alveolar epithelial cells [262,263]. MERS-CoV binds to dipeptidyl peptidase 4 (DPP4; also known as CD26) and exhibits wide tissue and cell tropism [264,265]. SARS-CoV-2 also uses ACE2 for entry [266] but has a broader cell tropism than SARS-CoV. In addition to ciliated epithelial cells and type II alveolar epithelial cells in the lungs, SARS-CoV-2 also infects intestinal epithelial cells and brain cells [267]. Zhu et al. showed that SARS-CoV-2 infection induced cytopathic effects and decreased TEER in organotypic human airway epithelial cultures. However, the effect of the virus on TJ and AJ was not examined in this study [76]. One study used a primary human bronchial epithelium model and showed that SARS-CoV-2 infection caused a transient decrease in TEER and increased paracellular permeability for dextran-FITC, where disrupted ZO-1 staining pattern and apoptosis were observed in infected epithelial cells [77]. The major cause of respiratory failure is damage to the epithelial–endothelial barrier of the alveolus [268,269]. Deinhardt-Emmer et al. elucidated that SARS-CoV-2 damaged the epithelial/endothelial barrier and induced robust immune reactions using a human chip model composed of epithelial, endothelial, and mononuclear cells. In this study, immunostaining of E-cadherin and scanning electron microscopy showed a disruption in epithelial barrier structure after viral infection. The investigators also noted that the epithelial/endothelial barrier permeability to FITC-dextran significantly increased [78].

Efforts to understand the impact of coronaviruses on the airway epithelial barrier have revealed some possible underlying mechanisms. In a recent publication, Rouaud et al. showed that ACE2 is localized and concentrated in the epithelial apical cell junction, which could facilitate the internalization of the virus across the airway epithelial cell barriers [270]. It has been reported that the SARS-CoV envelope (E) protein binds to human Proteins Associated with Lin Seven 1 (PALS1), which is a tight junction-associated protein crucial for the establishment and maintenance of epithelial polarity. The ectopic expression of the SARS-CoV E protein delayed the formation of TJs in kidney epithelial cell monolayers [75]. PALS1 was also shown to facilitate the intracellular traffic of E-cadherin in kidney epithelial cells, and the loss of PALS1 led to severe TJ and AJ disruption [271]. Interestingly, De Maio et al. predicted that the SARS-CoV-2 E protein binds more stably with PALS1 compared to SARS-CoV and could cause enhanced epithelial barrier disruption contributing to the pathogenesis of COVID-19 [272]. Moreover, there is evidence that the C-terminal domain of the SARS-CoV-2 E protein binds to the second PDZ domain of ZO-1 [273]. Although experimental evidence is still lacking, it is possible that SARS-CoV and the SARS-CoV-2 E protein bind to PALS1 and disrupt its trafficking to TJ and the proper trafficking of E-cadherin and ZO-1 to AJC in airway epithelial cells, resulting in the loss of barrier integrity. SARS-CoV-2 infection results in the downregulation of claudin-18a in induced pluripotent stem cell-derived alveolar epithelial type II cells, which might play a role in the loss of alveolar epithelial barrier function and pulmonary edema in COVID-19 patients [274]. Recent bioinformatics analysis identified the integral role of cell junction and cytoskeletal genes in COVID-19 and suggested their therapeutic potential [275]. SARS-CoV-2 infection also activates the release of proinflammatory cytokines, and their potential roles in barrier disruption have been reviewed by others [276]. Compared with the emerging studies on SARS-CoV and SARS-CoV-2, the impact of MERS on the airway epithelial barrier is not very well understood. As the COVID-19 pandemic is still an ongoing global public health problem, further investigation is needed to understand the pathogenesis of coronavirus and provide more information for drug development, especially the impact on epithelial barriers in both cultured cells and animal models.

## 4. Conclusions

With a large surface in direct exposure to the environment, the human respiratory epithelial barrier plays a crucial part in the host defense mechanisms against inhaled stimulants. Breaking the airway epithelial barrier is the stepping stone for environmental pathogens to establish infection in the respiratory tract. Mounting evidence has shown that impairing the AJC, a key contributing factor to cell–cell contact in the respiratory epithelium, is an important strategy used by bacteria and viruses to disrupt the integrity of the airway epithelial barrier. Here, we reviewed the recent findings regarding the impacts on AJC of a selected group of bacteria and respiratory viruses that impose substantial health and economic burdens. We highlighted the growing knowledge regarding the molecular mechanisms by which these microorganisms cause the dysregulation of AJCs and breach of the airway epithelial barrier. The reduced expression or mislocalization of TJ components ZO-1, occludin, and several claudin family members, as well as AJ proteins E-cadherin and β-catenin, have been highly implicated in pathogen-induced barrier dysfunction. Manipulating some of the underlying molecular pathways showed promising potential in reinstating epithelial barrier function. These advances present unique opportunities for the development of clinically relevant antimicrobial therapies. Targeting the host cells may also help to address the challenges of emerging pathogen resistance and the existence of multiple serological types.

Despite the growing knowledge of the mechanisms involved, their applications in mitigating epithelial barrier dysfunction in disease states require further and thorough examination. So far, most investigations have been conducted with in vitro models of the airway epithelial barrier. Future studies that examine the junctional structure and function in appropriate animal models or using 3D cell culture models such as organoids will be imperative to validate the mechanisms in vivo. Rapid advances in single-cell sequencing and spatial transcriptomics technologies will enable the molecular analysis of changes in the airway epithelial barrier with significantly improved time and spatial resolutions, which we foresee could accelerate the development of AJC-focused antimicrobial treatments. Identifying the molecular pathways that enhance AJC function will also be of earnest interest for its implications in the design of novel antimicrobial strategies.

## Figures and Tables

**Figure 1 pharmaceutics-14-02619-f001:**
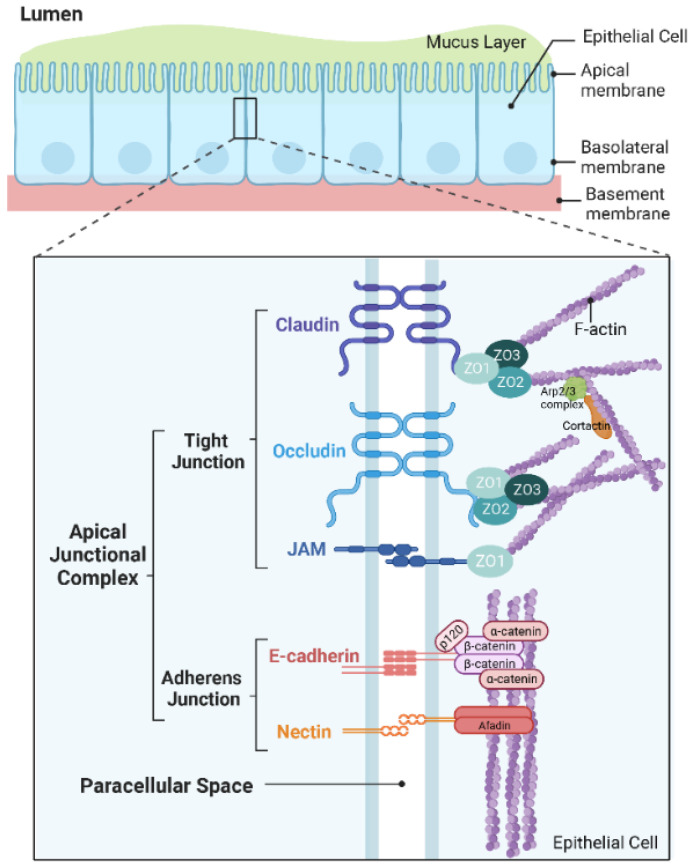
Schematic diagram illustrating the structure of apical junction complexes in the airway epithelium. In the human airway epithelium, the surface mucus layers and the physical barrier formed by airway epithelial cells serve as the first line of defense against the external environment. The apical junctional complexes (AJCs) between adjacent epithelial cells establish cell polarity and restrict epithelial permeability via maintaining cell–cell contact. AJCs are located on the top lateral membranes between neighboring cells and include tight junctions (TJs) and adherens junctions (AJs). The inset shows an enlarged illustration of several protein components of TJ and AJ. Claudin, occludin, and junctional adhesion molecule (JAM) are examples of TJ transmembrane proteins. E-cadherin and nectin are shown as examples of the AJ transmembrane proteins. These proteins interact with cytoplasmic adapter proteins, such as zonula occludens (ZO-1/2/3) and β-catenin, to connect to the prejunctional actin cytoskeleton. The assembly and remodeling of the filamentous actin (F-actin) network are critical for the epithelial barrier function and are regulated by key organizers such as cortactin and actin-related protein-2/3 (Arp2/3) complex. This image was created with BioRender.com.

**Figure 2 pharmaceutics-14-02619-f002:**
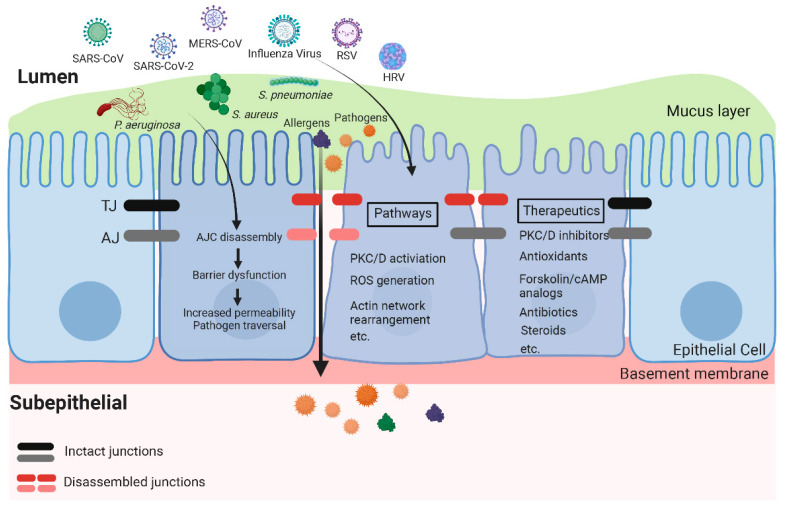
Disrupted epithelial barrier induced by invading microorganisms. Bacterial and viral infection can cause AJC disassembly and barrier disruption, the consequent increase in epithelial permeability will allow or exacerbate secondary invasions of allergens and pathogens into the subepithelial space. Multiple molecular pathways have been implicated in mediating or contributing to pathogen-induced AJC disassembly. Some examples are protein kinase C/D (PKC/D) activation, reactive oxygen species (ROS) generation, and rearrangement of the F-actin cytoskeleton. Therapeutics such as agonists and antagonists have been investigated and/or proposed to restore the epithelial barrier function caused in infected epithelial cells. Refer to Table 1; Table 2 for detailed information about the critical pathways and potential therapeutics. cAMP, cyclic adenosine monophosphate; HRV, human rhinovirus; IAV, influenza A virus; MERS-CoV, Middle East respiratory syndrome coronavirus; RSV, respiratory syncytial virus; SARS-CoV, severe acute respiratory syndrome coronavirus; SARS-CoV-2, severe acute respiratory syndrome coronavirus-2. This image was created with BioRender.com.

**Table 1 pharmaceutics-14-02619-t001:** Bacterial disruption of airway epithelial barrier and AJC.

Microorganism	Model	Impacts on Airway Epithelial Barrier Function	Impacts on Apical Junctional Complex (AJC)	Mechanisms	Potential Therapeutics	Ref
*Staphylococcus* *aureus (S. aureus)*	Primary epithelial cells from CRSwNP patients-ALI; *S. aureus* enterotoxin B	↓ TEER, ↑ permeability	↓ ZO-1 (IF), ↓ occludin (IF), ↓ phospho-occludin (WB), ↓ claudin-1(WB)	TLR2 pathway	TLR2 blocking antibody	[21]
	C57BL/6J mice; *S. aureus* enterotoxin B	↑ Permeability	↓ ZO-1 (mRNA), ↓ occludin (mRNA)			
	Primary HNECs-ALI; *S. aureus* conditioned media	↓ TEER	TJ separation (EM), discontinuous staining of ZO-1	-	-	[26]
	Primary HNECs-ALI; *S. aureus* extracellular proteases	↓ TEER, ↑ permeability	Discontinuous staining of ZO-1	-	-	[27]
	Human airway epithelial cells (H441)	↑ Permeability	↓ ZO-1 (IF), ↓ occludin (IF)	AMPK-PKCζ pathway	Metformin	[28]
	C57BL/6J mice; *S. aureus* α-toxin	↑ Permeability	↑ E-cadherin cleavage (WB)	ADAM10 activity	ADAM10 inhibition	[29]
	A549; *S. aureus* α-toxin	↓ Focal adhesion	-	FAK/Src/F-actin reorganization	-	[30]
	16HBE; *S. aureus* α-toxin	↑ Paracellular gaps	-	PAK/LIMK/cofilin/F-actin reorganization	-	[31]
*Streptococcus* pneumoniae *(S. pneumoniae)*	A549, 16HBE, C57BL/6J mice	Bacterial transmigration, ↑ permeability	-	-	IFN-β	[20]
	A549	-	↑ E-cadherin cleavage/degradation (WB)	ADAM10 activation by pneumolysin	ADAM10 inhibition	[29]
	16HBE, C57BL/6J mice	↓ TEER, ↑ permeability	↓ Claudin-7,10 (mRNA)	TLR/p38/TGF-β/Snail1	-	[32]
	Human respiratory tissues	-	TJ separation (EM)	-	-	[33]
	H292, BALB/c mice	-	↓ E-cadherin (IF)	-	-	[34]
	human lung tissue	-	(IF, WB) ↓ ZO-1, ↓ occludin, ↓ claudin-5, ↓ VE-cadherin	-	-	[35]
	A549	Bacterial transmigration	↑VE-cadherin cleavage/degradation (WB)	bacteria-bound plasmin	-	[36]
*Pseudomonas* *aeruginosa* *(P. aeruginosa)*	Calu3	↓ TEER, ↑ permeability	↓ occludin and claudin-1, ↑ cleavage of occludin (WB)	hyperglycemia	Metformin	[37]
	16HBE	↓ TEER, ↑ permeability	↓ ZO-1 (IF)	-	-	[38]
	C57BL/6 mice	↑ Permeability	-	-	-	
	Primary HNECs; *P. aeruginosa* elastase (PE)	↓ TEER (transient)	Transient ↓ claudin-1 and -4, occludin, and tricellulin (WB)	Decreased PAR-2; activation of PKC, MAPK, PI3K, p38 MAPK, JNK, COX-1 and -2, and NF-κB pathways	PAR-2 agonist; inhibitors for PKC, MAPK, PI3K, p38 MAPK, JNK, COX-1 and -2, and NF-κB pathways	[39]
	Calu3; *P. aeruginosa* elastase (PE)	↓ TEER	↓ localization of ZO-1 and occludin on membrane (IF, WB)	PKC signaling/ F-actin reorganization	PKC inhibitor	[40]
	Primary HNECs-ALI	↓ TEER, ↑ permeability	-	endotoxin rhamnolipids	-	[41]
	16HBE	↑ Permeability	Altered distribution of ZO-1 (IF), ezrin (IF), and occludin (IF, WB)	Type III toxins (ExoS, -T, and -Y. ExoS)	-	[42]
*Burkholderia*	16HBE	↓ TEER, ↑ permeability	↓ occludin (IF)	occludin dephosphorylation	-	[43]
	Human lung explant	↑ Permeability	-	-	-	[44]
	Primary Human AECs-ALI; *B. cepacia* BC-7	Invasion and destruction of epithelial cells (EM)	-	Biofilm-dependent, rearrangements of the actin cytoskeleton	-	[45]
	Primary Human AECs-ALI; *B. cepacia* HI2258	Invasion and destruction of epithelial cells (EM)	-	Biofilm-independent	-	
	Primary Human AECs-ALI; *B. cepacia* J-1	Invasion and destruction of epithelial cells (EM)	-	Biofilm-dependent and independent	-	
	Calu-3	↓ TEER	↓ ZO-1 and E-cadherin (IF)	-	-	[46]
	Primary HBECs-ALI, 16HBE	↓ TEER	Disrupted occludin (IF)	Increasing TNF-α cytokine	TNF-α neutralizing agent and steroids	[47]
	16HBE, CF cell line (CFBE41o−)	↓ TEER	↓ ZO-1 (IF, WB)	-	-	[48]
	16HBE, CF cell line (CFBE41o−)	↓ TEER, ↑ permeability	↓ ZO-1, occludin, and claudin-1 (WB)	-	-	[49]
*Haemophilus influenzae (H. influenzae)*	Primary human alveolar epithelial cells type II, A549	-	↓ E-cadherin (IF, WB, mRNA), ↓ ZO-1 (IF)	FGF2 upregulation and activation of mTOR pathway	rapamycin	[50]

List of abbreviations: 16HBE, 16HBE14o-human bronchial epithelial; ADAM10, metalloproteinase domain-containing protein 10; AECs, Airway epithelial cells; ALI, air–liquid interface; AMPK, AMP-activated protein kinase; CF, cystic fibrosis; COX-1 and -2, cyclooxygenase-1 and -2; CRSwNP, chronic rhinosinusitis with nasal polyps; EM, electron microscopy; F-actin, filamentous actin; FAK, focal adhesion kinase; HBECs, human bronchial epithelial cells; HNECs, human nasal epithelial cells; IF, immunofluorescence; JNK, c-Jun N-terminal kinase; MAPK, mitogen-activated protein kinase; NF-κB, nuclear factor-κB; PAR-2, protease-activated receptors; PI3K, phosphoinositide 3-kinase; PKC, protein kinase C; Src, steroid receptor coactivator; TEER, trans-epithelial electrical resistance; TGF-β, transforming growth factor-β; TLR, Toll-like receptor; TNF-α, tumor necrosis factor-α; TJ, tight junction; WB, Western blot; ZO, zonula occludens. ↓, decreased; ↑, increased; -, not determined.

**Table 2 pharmaceutics-14-02619-t002:** Virial disruption of airway epithelial barrier and AJC.

Microorganism	Model	Impacts on Airway Epithelial Barrier Function	Impacts on Apical Junctional Complex (AJC)	Mechanisms	Potential Therapeutics	Ref
Respiratory Syncytial Virus (RSV)	Primary mTECs -ALI	↓ TEER, ↑ permeability	disassembly of ZO-1, occludin, β-catenin (IF), ↑ E-cadherin cleavage (WB)	-	-	[15]
	C57BL/6J mice	↑ Permeability	↓ ZO-1 (IHC, WB), mislocalization of occludin (IHC), ↓ occludin (WB), ↓ claudin-1 (IHC, WB), ↑ claudin-2 (IHC, WB), ↑ E-cadherin cleavage (WB)	-	-	
	16HBE	↓ TEER, ↑ permeability	disassembly of ZO-1, occludin, E-cadherin, β-catenin (IF)	PKD activation/actin cytoskeletal remodeling	PKD inhibitors	[18]
	Primary NHBE cells-ALI, 16HBE	↓ TEER, ↑ permeability	disassembly of ZO-1, occludin, E-cadherin, β-catenin (IF)	-	Forskolin, cAMP analog	[51]
	Primary HBECs, A549	↓TEER, ↑ paracellular gaps	-	p38 MAPK activation/actin cytoskeletal remodeling	p38 MAPK inhibitor	[52]
	16HBE	↓ TEER, ↑ permeability	disassembly of ZO-1, occludin, E-cadherin, β-catenin (IF)	cortactin decrease/Rap1 inhibition/actin cytoskeletal remodeling	F-actin stabilizer, Rap1 activator	[53]
	BALB/c mice	↑ Permeability	↓ occludin and claudin-1 (mRNA)	-	-	[54]
	Primary HNECs	↑ TEER	↑ claudin-4 and occludin (IF, WB, mRNA)	TGF-β1/PKCδ/HIF-1α/NF-κB	-	[55]
	Primary HBECs	↓ TEER, ↑ paracellular gaps	-	inducing VEGF	VEGF antibody	[56]
Human Rhinovirus (HRV)	Primary human AECs-ALI	↓ TEER, ↑ permeability (transient)	↓ occludin, Crumbs3, and E-cadherin (IF, WB, mRNA)	EGFR activation and Snail increase	Snail inhibition	[57]
	Primary human AECs-ALI, 16HBE, calu-3, C57BL/6 mice	↓ TEER, ↑ permeability	dissociation of ZO-1 from TJ (IF, IHC, WB)	-	-	[58]
	Primary HNECs-ALI	↓ TEER	-	-	Betamethasone	[59]
	Primary NHBE cells-ALI	↓ TEER, ↑ permeability	disruption of ZO-1 and occludin (IF)	PGC-1α decrease	PGC-1α activator	[60]
	Primary human AECs-ALI, NuLi-1	↑ Permeability	↓ membrane ZO-1, occludin, and claudin-1 (In-Cell Western), ↓ ZO-1 (mRNA)	IL-15	-	[61]
	Primary HNECs-ALI	↓ TEER	↓ ZO-1, occludin, claudin-1, and E-cadherin (IF, WB, mRNA)	-	-	[62]
	Primary HNECs-ALI	-	↓ ZO-1, occludin, claudin-1, and E-cadherin (WB)	ROS-mediated phosphatases inhibition	NOX inhibitor	[63]
	Primary human AECs from asthmatic children-ALI	↓ TEER, ↑ permeability	↓ ZO-1 and occludin (In-Cell Western, IF), ↓ claudin-1 (In-Cell Western)	-	-	[64]
	Human precision-cut lung slices	-	↓ claudin-8 (mRNA)	-	-	[65]
	16HBE	↓ TEER, ↑ bacterial transmigration	↓ ZO-1, occludin (IF)	ROS generation	Rac1/NOX/NOX1 inhibitor, quercetin	[66]
	16HBE	↓ TEER, ↑ bacterial transmigration	↓ occludin (IF, WB)	mitochondrial ROS generation	antioxidant targeted to mitochondria	[67]
	Primary HNECs-ALI	-	Disruption of ZO-1, occludin, claudin-1, and E-cadherin (WB)	ROS generation	Ginsenoside Re	[68]
	Primary HNECs-ALI, RPMI 2650	-	Disruption of ZO-1, occludin, claudin-1, and E-cadherin (WB)	Akt/NF-κB and ERK1/2 activation	Wogonin	[69]
Influenza A Virus (IAV)	Coculture of human alveolar epithelial and endothelial cells; H1N1 and H5N1	↓ TEER, ↑ permeability	↓ AJC integrity (EM), ↓ JAM (IF), ↓ claudin-4 (IF)	-	-	[16]
	Primary NHBE cells-ALI; H1N1	↓ TEER, ↑ permeability	Disassembly of ZO-1 (IF)	-	-	[17]
	Primary human alveolar epithelial cells type II; H1N1	↓ TEER, ↑ permeability	-	-	-	[70]
	A549, C57BL/6J mice; H1N1	↓ TEER ↑ permeability	↓ ZO-1 (WB, IF) ↓ occludin (IF, WB) ↓ E-cadherin (WB)	MAPK/PI3K/Gli1/Snail activation	Gli1 Inhibitor	[71]
	A549, NL20, C57BL/6J mice; H5N1	↓ TEER	↓ ZO-1(WB), ↓ occludin (IF, WB), ↓ claudin-1 (IF, WB), ↓ E-cadherin (WB)	TAK1/Itch- mediated protein degradation	TAK1-Itch inhibition	[72]
Human Parainfluenza Virus (HPIV)	Primary human AECs-ALI; HPIV3 and HPIV5	↓ TEER, ↑ permeability	-	-	-	[73]
	A549; HPIV2	Cell-to-cell spread of HPIV2	↑ claudin-1 mRNA	HPIV2 V protein	Increasing claudin-1	[74]
Severe Acute Respiratory Syndrome (SARS-CoV)	MDCKII; SARS-E protein overexpression	↓ TEER	↓ E-cadherin and ZO-1(IF), delayed the formation of TJs (IF)	PALS1 Binding	-	[75]
Severe Acute Respiratory Syndrome 2 (SARS-CoV-2)	Organotypic human airway epithelial cultures-ALI	↓ TEER	-	-	-	[76]
	Reconstructed human bronchial epithelium-ALI	↓ TEER, ↑ permeability (transient)	Disrupted ZO-1 (IF)	-	-	[77]
	Chip model of human epithelial, endothelial, and mononuclear cells	↑ Permeability	Disrupted cell–cell contact (SEM), Disrupted E-cadherin (IF)	-	-	[78]

List of abbreviations: cAMP, cyclic adenosine monophosphate; EGFR, epidermal growth factor receptor; ERK1/2, extracellular signal-regulated protein kinase 1/2; HIF-1α, hypoxia-inducible factor 1-alpha; IHC, immunohistochemistry; IL, interleukin; mTECs, mouse tracheal epithelial cells; MDCKII, Madin–Darby Canine Kidney II epithelial cells; NHBE, normal human bronchial epithelial; NOX: NADPH oxidase; PGC-1α, peroxisome proliferator-activated receptor-gamma coactivator-1α; PKD, protein kinase D; ROS: reactive oxygen species; SEM, scanning electron microscopy; VEGF, vascular endothelial growth factor. ↓, decreased; ↑, increased; -, not determined.

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
