# Peer review of "Airway Epithelial Cell Junctions as Targets for Pathogens and Antimicrobial Therapy"

_pharmaceutics, 2022, doi:10.3390/pharmaceutics14122619_

Round 1
Reviewer 1 Report
1- try to increase the resolution of Figures 1 and 2
The paper was very well written and designed review paper by the authors of this review, was written in a sequential and elaborate manner, and quotes from the latest studies and literature published in various scientific journals.
Reviewer 2 Report
This is a comprehensive, clearly written and detailed review of the current knowledge about junctional targets of bacterial and viral pathogens in airway epithelial cells and tissues. There are two Tables, which provide an exhaustive description of the effects of a number of bacterial and viral pathogens on the barrier function TJ in different airway epithelial model systems, with correlated impacts on the expression and localization of junctional proteins, and mechanistic insights. Schematic Figures summarize some of the concepts described in the text. This review will be a very useful resource for researchers in the field of physiopathology of airway barriers, and potentially other barriers (intestine, kidney, etc).
The authors should however be more cautious about mechanistic conclusions when (as it is in most cases) there has not been a rigorous experimental demonstration of the involvement of one or another junctional protein in a certain phenotype/effect. It is hard to conclude that disrupted expression and/or localization of any specific junctional protein (with some exceptions) is causative of the impact of the pathogen on barrier function, due to functional redundancies. Correlation does always imply causation. For example, discontinuous staining of ZO-1 may be the result of cellular damage (caused for example by pore-forming toxins) and the effect on TEER or barrier to solutes may be independent of reduced ZO-1 labeling, but due to disruption of adhesion, etc. Similarly, occludin is not a major determinant of barrier function, therefore its mechanistic implication in altered barrier function cannot be inferred simply from disruption or decrease in its labeling. In summary, the authors should moderate and revise the sentences where they imply causation for observations that are more correctly only correlations.
Specific points:
1. Considering the content of the review, the title could be revised to include pathogens, e.g. “Airway epithelial cell junctions as targets for pathogens and antimicrobial therapy”
2. In the Introduction, first page, the sentence “Because of this “seal”, TJ maintain….” Should be revised to “In addition to this “seal”, TJ maintain…”. This is because there is actually no proof that claudin-based strands, that are responsible for barrier function, are directly implicated in the maintenance of cell polarity/fence function.
3. Immediately below, “TJ consist of” should be revised to “TJ comprise”, since TJ actually consist of not only transmembrane proteins, but also cytoplasmic and cytoskeletal components.
4. Introduction page 2. When mentioning Ig-like adhesion molecules, better to mention only JAM-A, since there is little known about the role of JAM-B and JAM-C in epithelial TJ.
5. Introduction page 2. In addition to the references 10, 11, the authors should cite Fanning et al 1998 JBC (The tight junction protein ZO-1establishes a link…) and Wittchen et al 1999 JBC (Protein interactions at the tight junction....).
6. Figure 1 should be revised to show, if possible, an organization of the actin filaments that are in a circumferential bundle parallel to the membrane at the level of the AJ, and a less bundled, and more branched actin filaments at the level of the TJ. This is because actin filaments are perpendicular to cell-cell contacts only in the very initial phases of junction formation, whereas in mature junctions they are organized in a much more stable, bundled fashion and parallel to the junctional plasma membrane, as detailed above.
7. In Fig. 1 the schematics for claudin and occludin should be switched, because it is occludin that has a longer cytoplasmic domain. In addition the ZO complex should be shown associated with that longer (C-terminal) domain, instead of the shorter N-terminal cytoplasmic tail.
8. In Fig. 1 the lateral membrane at the bottom should be shown straight and not curved, because in the present form it looks like that AJ is close to the basal part of the cell.
9. Page 10. The reference Shah et al Cell Rep 2018 (A Dock-and-Lock mechanism….) should be included, mentioning the mechanism through which PLEKHA7 promotes ADAM10-mediated toxicity of S. aureus, which implies two additional members of the complex (PDZD11 and Tspan33).
10. Page 16, paragraph “Evident disassembly…” instead of “emerging evidence indicates that changes in the molecular components of epithelial TJs and AJs are critical contributing factors”, say “emerging evidence indicates that changes in the molecular components of epithelial TJs and AJs occur during RSV infection”. As indicated above, correlation is not equivalent to causation, as correctly stated by the authors in the last sentence of the same paragraph.
11. Page 18, 2nd and third line, revise: ample evidence indicates that HRV infection-dependent impairment of the airway epithelial barrier correlates with disruption of the expression and localization of TJ/AJ proteins”.
12. Last paragraph of page 21 (continuing into page 22), cite the reference Rouaud et al 2022 (Cells Vol 11, issue4, “The ACE2 receptor for coronavirus entry…), with reference to the likely AJC localization of the ACE2 receptor in epithelial airway cells, based on its junctional localization in other epithelial cell types.
13. Miscellaneous:
a. Improve resolution of Fig. 1 and Figure 2. Some of the graphical items, for example actin filaments, appear fuzzy.
14. Typos:
a. Double . sign after first sentence in Fig. 1 legend.
b. Key organizer and not key organized (Legend Figure 1).
c. Table 2 Influenza A Virus occludin not occluding.
Reviewer 3 Report
This work is an endless compilation of citations. I find it useful to have them all together. As mentioned by the authors, the AJC is understudied, in spite of the 272 references the authors cite. I was surprised the authors did not mention the connections between claudin-1 and Hepatitis virus C (see here, https://pubmed.ncbi.nlm.nih.gov/17325668/) this article has been cited almost 200 times. The authors in reference 23 have access to a recent review describing said interaction with Hep C virus. JAM-A is also a port of entry for mammalian viruses, see your own reference 23 for more details. I do recognize that these examples might be outside the scope of the Lungs, as the review intends.
This work is extensive but not exhaustive. Much remains to understand, and in the future, this review should be very impactful.
